**∂ | Open Peer Review** | *Clinical Microbiology* | Research Article

# VesiX cetylpyridinium chloride is rapidly bactericidal and reduces uropathogenic *Escherichia coli* bladder epithelial cell invasion *in vitro*

Namrata V. Sawant,[1] Samuel S. Chang,[1] Krutika A. Pandit,[1] Prachi Khekare,[1] W. Randolph Warner,[2] Philippe E. Zimmern,[3] Nicole J. De Nisco[1,3]

**ABSTRACT** Management of urinary tract infection (UTI) in postmenopausal women can be challenging. The recent rise in resistance to most of the available oral antibiotic options together with high recurrence rate in postmenopausal women has further complicated treatment of UTI. As such, intravesical instillations of antibiotics like gentamicin are being investigated as an alternative to oral antibiotic therapies. This study evaluates the efficacy of the candidate intravesical therapeutic VesiX, a solution containing the cationic detergent Cetylpyridinium chloride, against a broad range of uropathogenic bacterial species clinically isolated from postmenopausal women with recurrent UTI (rUTI). We also evaluate the cytotoxicity of VesiX against cultured bladder epithelial cells and find that low concentrations of 0.0063% and 0.0125% provide significant bactericidal effect toward diverse bacterial species including uropathogenic *Escherichia coli* (UPEC), *Klebsiella pneumoniae*, *Enterococcus faecalis*, *Pseudomonas aeruginosa*, and *Proteus mirabilis* while minimizing cytotoxic effects against cultured 5637 bladder epithelial cells. Lastly, to begin to evaluate the potential utility of using VesiX in combination therapy with existing intravesical therapies for rUTI, we investigate the combined effects of VesiX and the intravesical antibiotic gentamicin. We find that VesiX and gentamicin are not antagonistic and are able to reduce levels of intracellular UPEC in cultured bladder epithelial cells.

**IMPORTANCE** When urinary tract infections (UTIs), which affect over 50% of women, become resistant to available antibiotic therapies dangerous complications like kidney infection and lethal sepsis can occur. New therapeutic paradigms are needed to expand our arsenal against these difficult to manage infections. Our study investigates VesiX, a Cetylpyridinium chloride (CPC)-based therapeutic, as a candidate broad-spectrum antimicrobial agent for use in bladder instillation therapy for antibiotic-resistant UTI. CPC is a cationic surfactant that is FDA-approved for use in mouthwashes and is used as a food additive but has not been extensively evaluated as a UTI therapeutic. Our study is the first to investigate its rapid bactericidal kinetics against diverse uropathogenic bacterial species isolated from postmenopausal women with recurrent UTI and host cytotoxicity. We also report that together with the FDA-approved bladder-instillation agent gentamicin, VesiX was able to significantly reduce intracellular populations of uropathogenic bacteria in cultured bladder epithelial cells.

**KEYWORDS** cetylpyridinium chloride, CPC, bladder instillation, urinary tract infection, UTI, recurrent urinary tract infection, rUTI, uropathogenic bacteria, uropathogens, gentamicin

Address correspondence to Nicole J. De Nisco, nicole.denisco@utdallas.edu.

W. Randolph Warner discloses an affiliation with US BioPharma.

See the funding table on p. 11.

Urinary tract infection (UTI) is among the most common adult bacterial infections affecting over 150 million people each year worldwide (1). UTI primarily affects women, and postmenopausal women are at a higher risk for developing recurrent UTI (rUTI), which is defined as two or more UTI episodes within 6 months or 3 or more episodes within 12 months (2, 3). The major etiological agent of rUTI is uropathogenic *Escherichia coli* (UPEC), followed by *Klebsiella pneumoniae*, *Staphylococcus saprophyticus*, *Proteus mirabilis*, *Pseudomonas aeruginosa*, and *Enterococcus faecalis* (4). These uropathogenic bacteria are highly adept at colonizing the urinary tract environment and evading host immune surveillance which allows them to persist in the urinary tract (5).

Oral antibiotic therapy is primarily used for UTI treatment (6). Unfortunately, in recent years, there has been a dramatic increase in the prevalence of antibiotic resistance among uropathogenic species as well as in the development of antibiotic allergies in patients undergoing long-term antimicrobial therapy (7). Use of broad-spectrum systemic antibiotics is also known to increase the risk of secondary infections such as *Clostridium difficile*-associated diarrhea (8). As per the WHO 2022 Global AMR surveillance report, high resistance rates to frequently used oral antibiotics such as ampicillin, trimethoprim/sulfamethoxazole, and fluoroquinolones were reported for *E. coli* isolates (9). Furthermore, UPEC, *E. faecalis*, *K. pneumoniae*, *P. mirabilis,* and *S. saprophyticus* have been shown to invade bladder epithelial cells to form intracellular bacterial communities (IBCs) and quiescent intracellular reservoirs (QIRs) (10–15). Such intracellular bacteria can persist within the bladder where they remain inaccessible to conventional antibiotic therapies (16–18) and act as reservoirs for rUTI (19, 20). As therapeutic options for rUTI become more limited, it is imperative to develop alternative strategies for their management.

A potential alternative for oral antibiotics is intravesical instillation, which involves administering antibiotic solutions directly into the bladder using a urinary catheter (21). Since intravesical instillation directly targets the bladder, a high concentration of an antibiotic can be used locally while avoiding the level of systemic side effects associated with oral antibiotics (22). Since the mid-1900s, various antimicrobial agents have been used as intravesical instillation solutions to treat UTI (23). Bladder irrigation with chlorhexidine alone and in combination with silver nitrate was shown to reduce the risk of catheter-associated UTI (24, 25) and decrease bacteriuria in patients with UTI (26). However, the presence of chlorhexidine in patient blood samples and severe erosive cystitis was reported with prolonged treatment (27, 28). Early case reports also indicate vancomycin and tobramycin as potential intravesical agents, but the clinical data on their efficacy are limited (29, 30). In 2013, Giua et al. reported use of colistin bladder instillations for UTI caused by multidrug-resistant *Acinetobacter* (31). However, this study was limited to three cases, with limited data on safety and optimum dosage. Several studies have also investigated the safety and efficacy of bladder instillation with gentamicin sulfate in the treatment of antibiotic-refractory rUTI in postmenopausal women and children (32–36). A study from 2017 demonstrated that intravesical gentamicin sulfate could lower symptomatic UTI and reduce use of oral antibiotics in neurogenic bladder patients with rUTI, however did not report reduction in rUTI episodes (37). Although no significant adverse effects of intravesical gentamicin sulfate have been reported, its inability to penetrate bladder epithelial cells could limit its efficacy in eliminating IBCs and QIRs (38). Despite the advantages offered by the bladder instillation, the studies on efficacy and safety of intravesical agents have been limited.

Cetylpyridinium chloride (CPC, CAS 123-03-5), a cationic quaternary ammonium compound with broad spectrum antimicrobial activity, may be a promising candidate antimicrobial agent for use in bladder instillation (39). The cationic hydrophilic region of CPC facilitates its binding to negatively charged bacterial cell membrane, thereby disrupting membrane integrity (40). This results in osmotic imbalance, leakage of cytoplasmic components, and eventually cell death (41–43). CPC has been reported to be safe with limited side effects for oral use even during pregnancy (44–46). The antibacterial activity of CPC in oral products has been extensively studied with clinical

studies reporting CPC's ability to effectively remove dental plaque and reduce gingivitis (40, 47, 48). It is commonly used as an active ingredient in over-the-counter oral hygiene products such as toothpastes, mouthwashes, throat sprays, nasal sprays, and lozenges (49, 50). Quisno et al. first studied the germicidal activity of CPC, wherein they reported CPC was bactericidal against a variety of Gram-positive and Gram-negative bacteria, fungi, and a flagellate (51). In 1941, another study reported CPC's sporicidal activity against *Clostridium perfringens*, *Clostridium sporogenes*, *Clostridium tetani*, *Bacillus anthracis*, and *Bacillus subtilis* (52). The antibacterial activity of CPC has also been demonstrated on biofilms of *Streptococcus mutans*, methicillin-resistant *Staphylococcus aureus*, *Actinomyces naeslundii*, and *Actinomyces odontolyticus* (53–55). CPC was also shown to impair the adhesion of the fungal pathogen, *Candida albicans*, to both biotic and abiotic surfaces (56). In addition to antibacterial and fungicidal activities, CPC also exhibits potent antiviral activity (39, 57–59).

Although the clinical efficacy and safety of CPC in oral hygiene is well documented, there have been limited studies on the activity of CPC against uropathogenic bacterial strains clinically isolated from the urine of UTI patients or its efficacy as an intravesical therapy for UTI. A study from 1945 suggested that CPC may be effective as a renal pelvic lavage to treat UTI (23). Furthermore, a recent limited trial of the efficacy of VesiX, a CPC-based antimicrobial agent, as an intravesical therapy for antibiotic-refractory rUTI found that VesiX effectively resolved symptomatic rUTI or reduced the antibiotic resistance profile of uropathogens in the majority of patients (60). To facilitate more focused and expanded trials for the use of CPC-based therapeutics in the management of antibiotic refractory rUTI, we aimed to define VesiX CPC therapeutic concentrations that would be broadly and rapidly bactericidal but not highly cytotoxic toward bladder epithelial cells and to examine potential synergy between VesiX and the FDA-approved intravesical agent gentamicin. We found that VesiX formulations with CPC concentrations of 0.0063% and 0.0125% showed greater than 3 $\log_{10}$ reduction in number of viable cells within 5 min of treatment for the majority of uropathogenic bacterial species tested. Notably, we observed that a combination treatment of 0.0004% CPC and 100 µg/mL GS significantly lowered uropathogenic *Escherichia coli* bladder epithelial cell invasion *in vitro*.

## RESULTS

### VesiX CPC solution is bactericidal against diverse species of uropathogenic bacteria

The antibacterial activity of the VesiX CPC solution was determined using broth microdilution assays against diverse species of uropathogenic bacteria including UPEC (*n* = 5), *K. pneumoniae* (*n* = 2), *E. faecalis* (*n* = 2), *P. aeruginosa* (*n* = 2), and *P. mirabilis* (*n* = 2) isolated from urine of postmenopausal women with active rUTI in standard bacteriologic MHB medium (Table 1). *E. coli* ATCC 25922 was used as a quality control strain (61). The minimum inhibitory concentration (MIC) and minimum bactericidal concentration (MBC) of VesiX for each bacterial strain are summarized in Table 2. Varying susceptibilities were observed for the tested species with MIC values ranging from 0.0002% to 0.0125%, and MBC values ranging from 0.0004% to 0.0125%. The MIC and MBC values for UPEC, *K. pneumoniae* and *E. faecalis* had were ≤0.0063%, whereas the MIC and MBC values for *P. aeruginosa* and *P. mirabilis* were ≤0.0125% (Table 2).

Furthermore, the host microenvironment has been shown to affect antibiotic efficacy (68), and testing in media which mimics the *in vivo* physiological conditions of the host may increase the accuracy of MIC prediction (69). To account for this, we also determined the MIC and MBC of VesiX CPC in pooled human urine supplemented with 30% (vol/vol) MHB (69) (Table 3). A twofold increase in MIC and MBC of VesiX CPC was observed for UPEC UTI89, *E. faecalis* ATCC 29212, *P. mirabilis* H14320, and *K. pneumoniae* TOP52 1721 in pooled human urine compared to MHB medium. The MBC of VesiX CPC increased fourfold for *P. aeruginosa* Cntrl_4A in pooled human urine compared to MHB. Whereas the MIC and MBC of VesiX CPC in pooled human urine were reduced by onefold in

**TABLE 1** Strain list of bacteria used in this study

| Strain ID | Species | Source |
|---|---|---|
| ATCC 25922 | *E. coli* | ATCC |
| UT189 | UPEC | Mulvey et al. (62) |
| PF2 | UPEC | This study |
| PF19 | UPEC | This study |
| PNK006 | UPEC | De Nisco et al. (63) |
| PNK007 | UPEC | De Nisco et al. (63) |
| ATCC 29212 | *E. faecalis* | ATCC |
| PF13 | *E. faecalis* | Sharon et al. (64) |
| Pre_VSX1 UK1 | *E. faecalis* | Zimmern et al. (60) |
| TOP52 1721 | *K. pneumoniae* | Rosen et al. (13) |
| PF18-2 UK1 | *K. pneumoniae* | This study |
| KpPF25 | *K. pneumoniae* | Sharon et al. (65) |
| PF18-2 UK4 | *P. aeruginosa* | This study |
| Cntrl_4A | *P. aeruginosa* | This study |
| HI4320 | *P. mirabilis* | Pearson et al. (66) |
| PM1668 | *P. mirabilis* | Nguyen et al. (67) |
| PM123 | *P. mirabilis* | Nguyen et al. (67) |

comparison to MHB for *E. coli* ATCC 25922. Also, the ratio of MBC to MIC of the VesiX CPC solution in both MHB (Table 2) and pooled human urine (Table 3) for all test strains was <4, confirming its bactericidal activity.

## Time-kill kinetic profile of VesiX CPC solution against uropathogenic bacteria

To evaluate the rate at which VesiX CPC solution effectively kills diverse species of uropathogenic bacteria, time-kill curves were constructed for early log-phase cultures of different bacterial species growing in the presence of CPC (Fig. 1). Based on the results of broth microdilution assays, 0.0063% and 0.0125% CPC were used as test concentrations (Fig. 1A and B). The average $\log_{10}$ reduction in a number of viable cells within 15 min of exposure to test concentrations of CPC, as presented in Fig. 1C, indicated that CPC exhibited rapid bactericidal activity. After 5 min of exposure to 0.0063% CPC, UPEC, *K. pneumoniae* and *E. faecalis* showed a greater than 3 $\log_{10}$ reduction in the viable cell count relative to the initial inoculum. The reduction in viable cell count of *P. mirabilis* was also greater than 3 $\log_{10}$ within 5 min of exposure to 0.0125% CPC. However, *P.*

**TABLE 2** Minimum inhibitory and bactericidal concentrations of VesiX CPC and gentamicin in MHB medium[a]

| Strain ID | Species | VesiX CPC | | | Gentamicin | | |
|---|---|---|---|---|---|---|---|
| | | MIC (%) | MBC (%) | MBC/MIC | MIC (µg/mL) | MBC (µg/mL) | MBC/MIC |
| ATCC 25922 | *E. coli* | 0.0008 | 0.0016 | 2 | 1.875 | 1.875 | 1 |
| UTI89 | *E. coli* | 0.0016 | 0.0016 | 1 | 1.875 | 1.875 | 1 |
| PF2 | *E. coli* | 0.0016 | 0.0008 | 0.5 | 1.875 | 1.875 | 1 |
| PNK006 | *E. coli* | 0.0008 | 0.0008 | 1 | 3.75 | 3.75 | 1 |
| PNK007 | *E. coli* | 0.0016 | 0.0016 | 1 | 1.875 | 1.875 | 1 |
| PF19 | *E. coli* | 0.0008 | 0.0016 | 2 | 1.875 | 1.875 | 1 |
| PF13 | *E. faecalis* | 0.0002 | 0.0004 | 2 | 7.5 | 15 | 2 |
| Pre_VSX1 UK1 | *E. faecalis* | 0.0002 | 0.0004 | 2 | 15 | 30 | 2 |
| PF18-2 UK1 | *K. pneumoniae* | 0.0031 | 0.0063 | 2.03 | <0.46875 | <0.46875 | 1 |
| KpPF25 | *K. pneumoniae* | 0.0016 | 0.0016 | 1 | 0.9375 | 0.9375 | 1 |
| PF18-2 UK4 | *P. aeruginosa* | 0.0125 | 0.0125 | 1 | 3.75 | 3.75 | 1 |
| Cntrl_4A | *P. aeruginosa* | 0.0125 | 0.0125 | 1 | 3.75 | 3.75 | 1 |
| PM1668 | *P. mirabilis* | 0.0063 | 0.0125 | 1.98 | 480 | 480 | 1 |
| PM123 | *P. mirabilis* | 0.0063 | 0.0125 | 1.98 | 240 | 240 | 1 |

[a]MIC, minimum inhibitory concentration; MBC, minimum bactericidal concentration.

**TABLE 3** Minimum inhibitory and bactericidal concentrations of VesiX CPC in pooled human urine supplemented with 30% (vol/vol) MHB medium[a]

| Strain ID | Species | VesiX CPC | | | | | |
| --- | --- | --- | --- | --- | --- | --- | --- |
| | | MHB | | | Pooled human urine | | |
| | | MIC (%) | MBC (%) | MBC/MIC | MIC (%) | MBC (&) | MBC/MIC |
| ATCC 25922 | *E. coli* | 0.0008 | 0.0016 | 2 | 0.0004 | 0.0004 | 1 |
| UTI89 | *E. coli* | 0.0016 | 0.0016 | 1 | 0.0031 | 0.0031 | 1 |
| ATCC 29212 | *E. faecalis* | 0.0002 | 0.0002 | 1 | 0.0004 | 0.0004 | 1 |
| TOP52 1721 | *K. pneumoniae* | 0.0016 | 0.0016 | 1 | 0.0031 | 0.0031 | 1 |
| Cntrl_4A | *P. aeruginosa* | 0.0125 | 0.0125 | 1 | 0.025 | 0.05 | 2 |
| H14320 | *P. mirabilis* | 0.0063 | 0.0063 | 1 | 0.0063 | 0.0125 | 2 |

[a]MIC, minimum inhibitory concentration; MBC, minimum bactericidal concentration.

*aeruginosa* was more resistant to CPC with only a 2.04 $\log_{10}$ reduction in number of viable cells within 15 min of exposure to 0.0125% CPC.

### *In vitro* cytotoxicity of VesiX CPC solution toward cultured human bladder epithelial cells

To evaluate potential off-target cytotoxic effects of the VesiX CPC solution, cytotoxicity against human-cultured bladder epithelial cells was determined using lactate dehydrogenase (LDH) assay. Time intervals were selected for their direct relevance to the timing of intravesical instillations which do not typically exceed 45 min (70). Our data indicated that cytotoxicity toward the human urinary bladder carcinoma cell line 5637 was <20% within 15 min of exposure to 0.0008%, 0.0031%, 0.0063%, and 0.0125% VesiX (Fig.

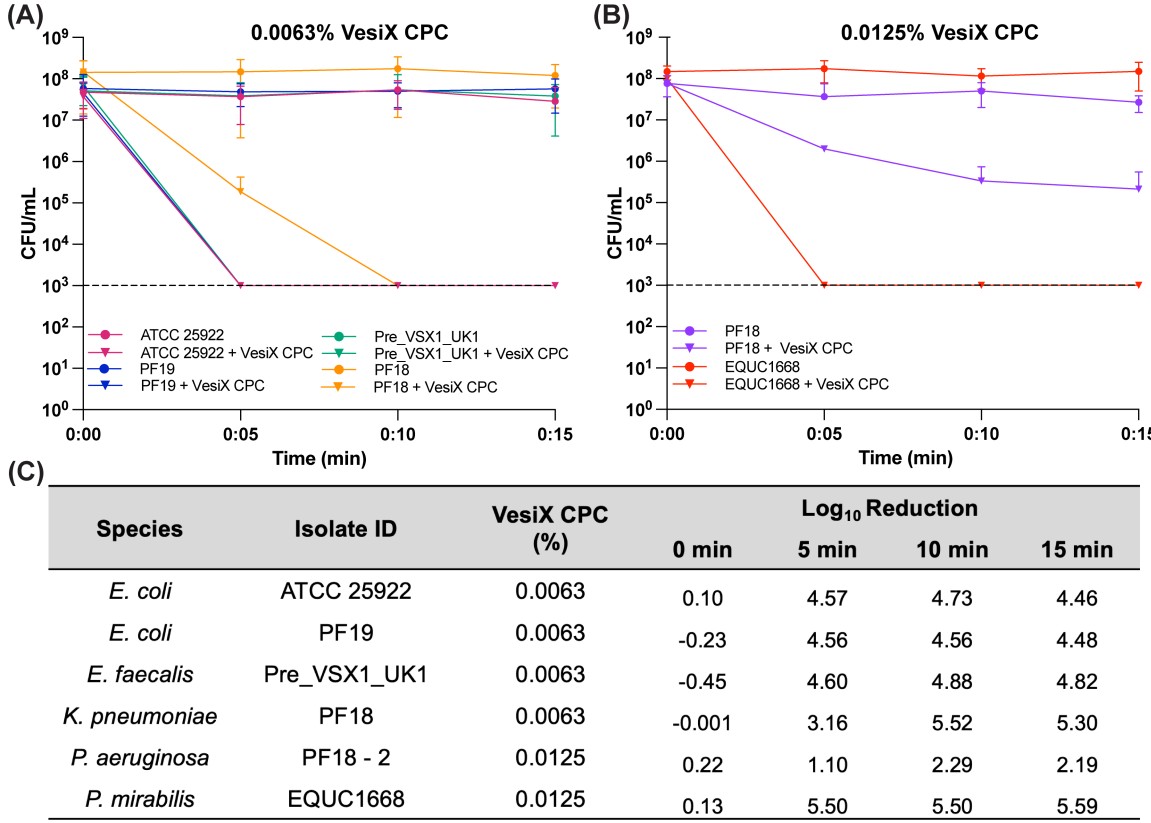

**FIG 1** Time-kill assays of uropathogens treated with 0.0063% and 0.0125% VesiX CPC. Growth curves of uropathogenic bacteria grown in Mueller–Hinton broth (dot) and in the presence of (A) 0.0063% and (B) 0.0125% VesiX CPC (inverted triangle). Data are presented as mean ± SD of three biological replicates. Dotted line represents the limit of detection. (C) $\log_{10}$ reduction of viable bacteria after treatment with 0.0063% and 0.0125% VesiX CPC. Data are presented as mean ± SD of three biological replicates.

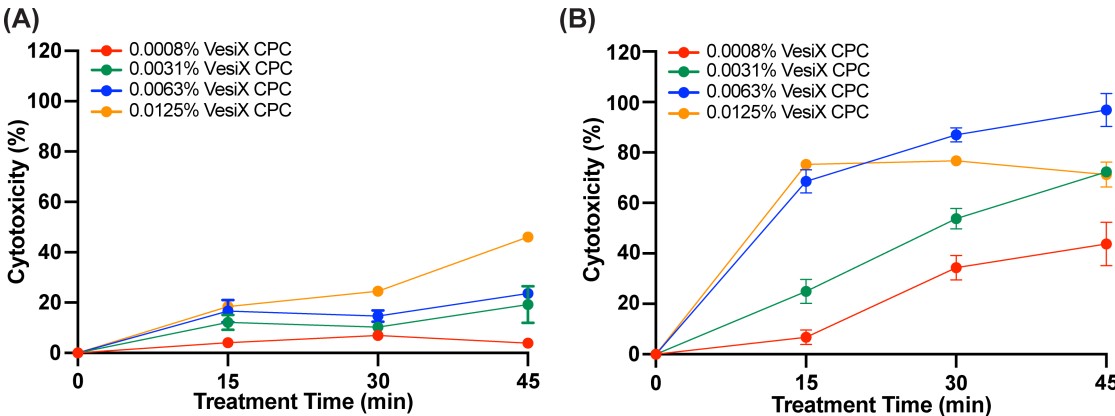

**FIG 2** *In vitro* cytotoxicity of VesiX CPC. Bladder epithelial cells (A) 5637 and (B) T24 treated with 0.0008%, 0.0031%, 0.0063%, and 0.0125% VesiX CPC for 45 min at 37°C in a humidified incubator with 5% $CO_2$. Percent cytotoxicity based on the quantification of LDH enzymatic activity in culture supernatant. Data are presented as mean ± SD of three biological replicates.

2A). After 45 min, percent cytotoxicity against 5637 cells remained low with 0.0008%–0.0063% VesiX but increased to 45% in cells treated with the higher dose of 0.0125% VesiX. Higher VesiX cytotoxicity was observed toward T24 cells relative to 5637 cells. In T24 cells, <20% cytotoxicity was observed within 15 min of exposure to 0.0008% VesiX CPC, while 0.0031%, 0.0063%, and 0.0125% VesiX CPC concentrations resulted in >20% cytotoxicity relative to Triton X-100 (Fig. 2B). However, the increased sensitivity of T24 cells was expected because unlike 5637 cells, they do not express uroplakins which help to limit cellular permeability (71).

## VesiX CPC and gentamicin co-treatment reduces intracellular UPEC in cultured bladder epithelial cells

Commonly used intravesical antibiotics like gentamicin may be effective in removing planktonic bacteria, but do not penetrate cellular membranes, so intracellular bacteria or bacteria embedded in the bladder wall may persist and cause rUTI (18). Because CPC can disrupt cell membranes, a combination intravesical treatment using both a CPC-based therapeutic and gentamicin may be more effective in reducing loads of intracellular or tissue-resident bacteria. We first sought to identify potential negative or positive interactions between VesiX CPC solution and gentamicin *in vitro* via checkerboard assay. UTI89 was used as a representative UPEC strain and ATCC 25922 was used as a quality control reference strain. Checkerboard assays revealed that the combination of VesiX CPC and gentamicin resulted in indifferent effect over the growth inhibition UTI89 and ATCC 25922 (ΣFIC index = 3, Fig. 3A) compared to the individual MICs. Since VesiX and gentamicin were not antagonistic, next we tested the efficacy of 0.004% CPC and 100 µg/mL gentamicin in eliminating intracellular UTI89 in 5637 bladder epithelial cells by invasion assay. The combination of VesiX CPC and gentamicin reduced the average CFU/mL of viable intracellular UTI89 by 1 $\log_{10}$ (90% reduction) in cultured bladder epithelial cells compared to gentamicin alone (Fig. 3B and C).

## DISCUSSION

Recurrent UTI is a major healthcare concern in postmenopausal women. Moreover, with chronic and rUTI, antibiotic refractory infections can be particularly challenging to manage (72). In such situations, intravesical antibiotic instillation may overcome some of the disadvantages of oral antibiotic therapy such as systemic side effects or limited effective dose (22). With intravesical antibiotic instillation showing promise for advanced UTI management, it is imperative to study antimicrobial agents that could be potentially used for intravesical instillation. In a recent limited trial, our group has shown that VesiX, a CPC-based antimicrobial agent, may be a safe therapeutic option for

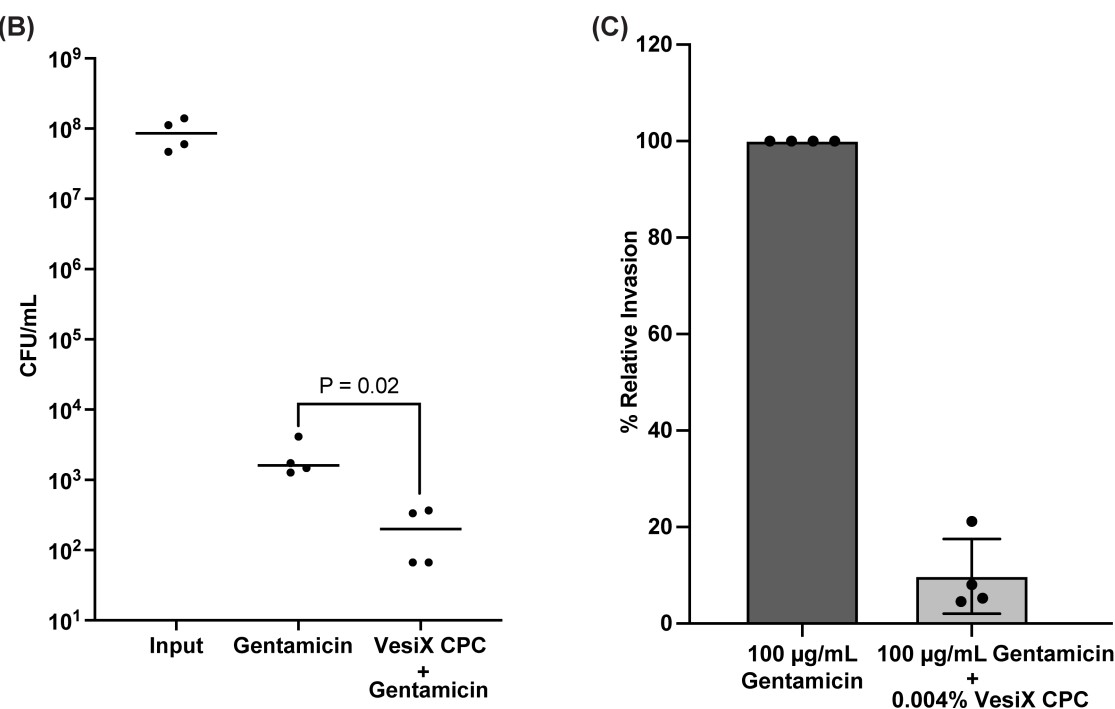

**(A)**

| Strain ID | Individual MIC | | Combination MIC | | FIC | | FIC Index | Interpretation |
|---|---|---|---|---|---|---|---|---|
| | Gentamicin | VesiX CPC | Gentamicin | VesiX CPC | Gentamicin | VesiX CPC | | |
| ATCC 25922 | 1.875 | 0.0008 | 1.875 | 0.0008 | 1 | 2 | 3 | Indifference |
| UTI89 | 1.875 | 0.0125 | 1.875 | 0.0008 | 1 | 2 | 3 | Indifference |

**FIG 3** Combination effect of VesiX CPC and gentamicin on intracellular UTI89. (A) Fractional Inhibitory Concentration (FIC) index for VesiX CPC and gentamicin in combination. ΣFIC index ≤0.5, synergistic; ΣFIC index >0.5 and ≤1.0, additive; ΣFIC index >1.0 and ≤4, indifferent; and ΣFIC index >4, antagonistic. Data are presented as mean of three biological replicates. (B) Colony-forming unit (CFU) enumeration and percent relative invasion (C) of intracellular UTI89 from 5637 cells treated with 0.0004% VesiX CPC and 100 µg/mL gentamicin in combination relative to percent invasion of UTI89 treated with 100 µg/mL gentamicin alone (two-tailed unpaired $t$-test).

the treatment of antibiotic-refractory rUTI in postmenopausal women (60). The present study extends our efforts on designing an optimal VesiX instillation treatment strategy for women with rUTI. In this study, we set out to determine the optimal time course and concentration for bactericidal action of VesiX CPC against diverse uropathogenic bacteria with minimal host cytotoxicity. To do so, we assessed the bactericidal activity of VesiX against UPEC, *K. pneumoniae*, *E. faecalis*, *P. aeruginosa,* and *P. mirabilis* isolated from urine of postmenopausal women with active rUTI. Our data indicate that uropathogenic species vary in susceptibility to VesiX. The lowest resistance was observed among *E. faecalis* (MIC and MBC ≤0.004%) and UPEC (MIC and MBC ≤0.0016%) isolates. The highest resistance to VesiX was found among *P. aeruginosa* and *P. mirabilis isolates* (MIC and MBC ≤0.05%), followed by *K. pneumoniae* (MIC and MBC ≤0.0063%). This observation supports the findings of Masadeh et al. that *P. aeruginosa* and *P. mirabilis* species exhibit resistance to CPC (73). Furthermore, the time-kill kinetics profile of CPC-based VesiX formulations showed greater than 3 $\log_{10}$ reduction in number of viable cells within 5 min for UPEC, *K. pneumoniae*, *E. faecalis,* and *P. mirabilis* while *P. aeruginosa* showed only a 2.04 $\log_{10}$ reduction even at the 15 min time-point. Future work should include mechanistic studies of the higher tolerance of *P. aeruginosa* and *P. mirabilis* to VesiX CPC. Gram negative bacteria such as *P. aeruginosa, P. mirabilis,* and *K. pneumoniae* can

exhibit higher resistance and tolerance to cell envelope-targeting antimicrobials due to their outer membrane modifications and ability to produce diverse capsules and biofilms (74–76).

From a safety standpoint, host cytotoxicity should be considered when designing an optimal intravesical antibiotic instillation strategy. Ideally, an instillation agent should kill or inhibit the growth of pathogenic bacteria in the bladder, without causing undue toxicity to the host (77). Our *in vitro* data demonstrate less than 20% cytotoxicity in the non-muscle-invasive bladder carcinoma cell line 5637 within 15 min of exposure to VesiX formulations with CPC concentration has high as 0.0063% and 0.0125%. A limitation of these findings is that 5637 cells have mutations in TP53, RB1, and ERBB2 genes, while T24 cells also have mutated TP53 and HRAS genes (Cancer Cell Line Encyclopedia; RRID:SCR_013836). These oncogenic alterations are known to activate pro-survival pathways, inhibiting cell death and could potentially impact the accurate prediction of cytotoxicity (78–81). To overcome this limitation, additional studies with normal bladder epithelial cells could be performed to further assess the cytotoxic effect of VesiX CPC. Additionally, even though limited human data suggest that higher concentrations of CPC (i.e., 0.05%) are well tolerated in intravesical therapy, an important next step will be to investigate *in vivo* toxicity of intravesical VesiX in appropriate animal models (60).

While intravesical gentamicin therapy is currently used in patients suffering from rUTI or UTI that are resistant to oral antibiotics, the inability of gentamicin to penetrate epithelial cells may limit its ability to clear intracellular bacteria (38). We, therefore, hypothesized that a combination intravesical treatment using both CPC and gentamicin may be more effective against intracellular bacteria because CPC, in addition to itself being bactericidal, may increase the permeability of bladder epithelial cells to gentamicin. Permeation enhancers such as surfactants and bile salts are often used to enhance drug delivery. Likewise, CPC was reportedly used as a permeation enhancer to effectively boost epithelial cell permeation of lorazepam (82). To investigate whether combination intravesical treatment of CPC with gentamicin may be more effective than GS used alone, we used an *in vitro* model simulating UPEC infection and invasion of bladder epithelial cells. We then showed that combination of 0.004% CPC and 100 µg/mL gentamicin reduced intracellular populations UTI89 in cultured bladder epithelial cells *in vitro*.

Taken together, our results suggest that a combination intravesical treatment of CPC with gentamicin may be able to eliminate of intracellular UPEC that could seed future episodes of rUTI. One limitation of this study is that it only assesses the antimicrobial efficacy of VesiX CPC against intracellular uropathogenic *Escherichia coli* and not other uropathogenic species that can also form intracellular reservoirs. Future studies can be directed toward the evaluation of VesiX CPC activity against intracellular populations of other invasive uropathogenic species such as *E. faecalis*, *K. pneumoniae*, *P. mirabilis,* and *S. saprophyticus* either *in vitro* or *in vivo* (12–15). A final limitation of this study is that the ability of VesiX CPC to reduce intracellular UPEC was only shown *in vitro* in cultured bladder epithelial cells. In future studies, it will be critical to evaluate the efficacy of VesiX CPC in eliminating intracellular bacterial communities and quiescent intracellular reservoirs in mouse models of rUTI.

## MATERIALS AND METHODS

### Bacterial strains and culture conditions

UTI89 is a uropathogenic *Escherichia coli* (UPEC) strain isolated from a patient with acute cystitis (62). *E. coli* (Migula) Castellani and Chalmers ATCC 25922 (ATCC, USA), *Enterococcus faecalis* (Andrewes and Horder) Schleifer and Kilpper-Balz ATCC 29212 (ATCC, USA), *Proteus mirabilis* HI4320 (66), and *Klebsiella pneumoniae* TOP52 1721 (13) were used as a quality control reference strains for antimicrobial susceptibility testing (61). The remaining bacterial strains used in this study, including UPEC (PF5, PF19, PNK006, PNK007), *Klebsiella pneumoniae* (PF18-2 UK1, KpPF25), *Enterococcus faecalis* (PF13, Pre_VSX1 UK1), *Pseudomonas aeruginosa* (PF18-2 UK4, Cntrl_4A), and *Proteus*

*mirabilis* (PM1668, PM123) isolated from urine of consenting postmenopausal women with active rUTI (Table 1) following IRB approval (STU 082010-016). In brief, 100 µL of urine was plated on a CHROMagar Orientation and incubated in ambient conditions at 37°C for 18–24 h for colony-forming unit (CFU) enumeration and differentiation of bacterial species. Well-isolated colonies with distinct morphologies were grown in Brain Heart Infusion (BHI) broth. 16S rRNA gene sequencing was used to confirm species identity.

## Eukaryotic cell culture

The human urinary bladder carcinoma cells 5637 (male) and T24 (female) used in this study were obtained from the American Type Culture Collection (Manassas, VA 20108, USA). The cells were cultured in RPMI 1640 medium with L-glutamine supplemented with 10% Fetal Bovine Serum, 1% (vol/vol) antibiotic (100 U/mL Penicillin and 0.1 mg/mL Streptomycin, Sigma-Aldrich) at 37°C in a humidified incubator with 5% $CO_2$.

## Antimicrobial agents

VesiX CPC solution [0.05% Cetylpyridinium chloride monohydrate, 1.5% (wt/vol) sodium chloride] was obtained from US BioPharma LLC (Punta Gorda, FL, USA). Gentamicin sulfate (GS) salt (potency ≥590 µg Gentamicin base per mg) from (Sigma-Aldrich, St. Louis, MO, USA) was used.

## Determination of minimum inhibitory and bactericidal concentrations

Broth microdilution method was used to determine the MICs and MBCs of CPC-based VesiX in Mueller–Hinton broth (MHB) and pooled human urine (Cat# IRHUURE; Innovative Research) according to the CLSI and EUCAST guidelines (83, 84). The pooled human urine was supplemented with 30% (vol/vol) MHB to support sufficient bacterial growth (69). In brief, serial dilutions of VesiX CPC solution were inoculated with a fixed number of test bacteria in a 96-well microtiter plate, and growth inhibition was measured. *Escherichia coli* ATCC 25922, *Enterococcus faecalis* ATCC 29212, *Proteus mirabilis* HI4320, and *Klebsiella pneumoniae* TOP52 1721 were used as quality control strains. Two-fold dilutions of VesiX prepared in MHB and pooled human urine ranging from 0.0000244% to 0.05% were dispensed in the respective wells of a microtiter plate. Growth control and sterility control wells were included. Each well containing VesiX and growth control well was inoculated with test bacteria to a final concentration of $4 \times 10^5$ CFU/mL. After 16–20 h of static incubation at 37°C, optical density ($OD_{600}$) was measured to determine growth inhibition. The lowest concentration of VesiX at which no bacterial growth was observed ($OD_{600} < 0.1$) was considered the MIC of VesiX for the test bacteria. For MBC determination, 10 µL from the MIC well and the two wells preceding the MIC well were spotted onto Mueller–Hinton agar (MHA) plates. Colonies were counted after 16–20 h incubation at 37°C. The dilution which resulted in ≥3 $\log_{10}$ reduction of viable bacteria was recorded as the MBC of VesiX for the test bacteria. Furthermore, the effect of VesiX was considered bactericidal if the ratio of MBC to MIC was ≤4, and bacteriostatic if the ratio of MBC to MIC was >4 (85). All assays were performed in biological triplicate and data were expressed as mean values.

## Time-kill kinetic assay

0.0063% and 0.0125% of VesiX CPC solution were inoculated with early log-phase cultures of test bacteria, normalized to $OD_{600}$ of 0.01 and incubated at 37°C, shaking for 1 h. At defined time intervals (0, 5, 10, and 15 min), 10 µL aliquots was taken, serially diluted, and spotted onto MH agar plates for CFU enumeration. After 16–20 h static incubation at 37°C, ≥3 $\log_{10}$ reduction in CFU/mL compared with the initial inoculum was considered bactericidal for VesiX. Time-kill curves were constructed by plotting the $\log_{10}$ CFU/mL versus time, and the change in bacterial population within 15 min of exposure to 0.0063% and 0.0125% of VesiX was determined. The experiments were

performed in biological triplicate. Statistical analysis was performed using GraphPad Prism software (GraphPad Software, La Jolla, CA, USA).

## Cytotoxicity assay

Lactate dehydrogenase (LDH) cytotoxicity detection kit (Takara Bio Inc.) was used to assess the viability of human urinary bladder epithelial carcinoma cells 5637 and T24 treated with four dilutions of VesiX CPC solution (0.0008%, 0.0031%, 0.0063%, and 0.0125%) prepared in colorless RPMI 1640 medium. Although McCoy's medium may be the optimal medium for T24 culture, as RPMI is also able to support robust T24 gtowth, RPMI was used for all experiments so that culture medium would be standardized between experiments. $1.5 \times 10^5$ cells/mL treated with the various VesiX dilutions were incubated for 15, 30, and 45 min at 37°C in a humidified incubator with 5% $CO_2$. Three sets of replicates for each condition were used, including the positive (10% Triton X-100) and negative (colorless RPMI 1640) controls. At the end of each time-point, 200 µL of supernatant from each well was transferred to a 96-well plate. The plate was then centrifuged for 5 min at 1,000 RPM. One hundred microliters of the supernatant from each well was then transferred to another 96-well plate. LDH detection kit reagents were prepared according to the manufacturer's protocol, and 100 µL of the reagent mix was added to the assay plate. At set time intervals, absorbance was measured at 490 nm providing a colorimetric measure of cell cytotoxicity. The acceptable level of cytotoxicity was determined as per ISO 10993-5:2009. Assays were performed in biological triplicate.

## Checkerboard assay

A two-dimensional broth microdilution checkerboard method was used to evaluate the activity of VesiX CPC solution and Gentamicin in combination against the diverse uropathogenic bacteria. In a 96-well microtiter plate, VesiX was serially diluted (0.00005%–0.0125%) along the X-axis, and Gentamicin was serially diluted along the Y-axis (0.063–4 µg/mL) resulting in each well containing a unique combination of VesiX and Gentamicin. Each well was inoculated with $4 \times 10^5$ CFUs of the test bacteria, and optical density was read at 600 nm after 16–20 h of static incubation at 37°C. Sterility and growth control wells were included in each assay. The total Fractional Inhibitory Concentration (ΣFIC) index was calculated to evaluate the interaction between CPC and Gentamicin using the formula: ΣFIC = FIC of agent A + FIC of agent B where FIC = MIC of agent in combination/MIC of agent alone (86). The results were interpreted as follows: ΣFIC index ≤0.5, synergistic; ΣFIC index >0.5 and ≤1.0, additive; ΣFIC index >1.0 and ≤4, indifferent; and ΣFIC index >4, antagonistic (87).

## Invasion assay

Gentamicin protection assay, as described previously (88), was used to study the effect of VesiX CPC solution alone and in combination with Gentamicin on bacterial invasion of the human urinary bladder carcinoma cells 5637. In brief, 5637 cells were seeded at $1.5 \times 10^5$ cells/mL in 6-well plates for 24 h before infection. Cells were infected with UPEC UTI89 at a multiplicity of infection (MOI) of 10 bacteria/cell, centrifuged at 600×$g$ for 8 min, and incubated at 37°C/5% $CO_2$ for 2 h. Cells in the input (control) wells were lysed with 0.3% Triton X-100, serially diluted, and spotted on Lysogeny Broth (LB) agar plates for bacterial CFU/mL enumeration. VesiX (0.004%) alone or with Gentamicin (100 µg/mL) was added to the output (test) wells. After 2 h incubation, cells were lysed with 0.3% Triton X-100. The lysate was serially diluted and spotted on LB agar plates for viable bacterial CFU/mL enumeration.

## ACKNOWLEDGMENTS

The authors would like to thank all the members of the De Nisco lab for their input throughout this study.

This study was supported by the National Institutes of Health 1R01DK131267-01 (N.J.D.) and by the Felecia and John Cain Distinguished Chair in Women's Health (P.E.Z.).

Conceptualization, N.V.S., P.E.Z., and N.J.D.; data curation, N.V.S., S.S.C., and N.J.D.; formal analysis, N.V.S., S.S.C., and N.J.D.; funding acquisition, P.E.Z. and N.J.D.; investigation, N.V.S., S.S.C., K.A.P., and P.K.; methodology, N.V.S., S.S.C., P.E.Z., and N.J.D.; project administration, N.V.S. and N.J.D.; resources, W.R.W., P.E.Z., and N.J.D.; supervision, N.J.D.; validation, N.V.S. and S.S.C.; visualization, N.V.S., S.S.C., and N.J.D.; writing – original draft, N.V.S., P.E.Z., and N.J.D.

## AUTHOR AFFILIATIONS

[1]Department of Biological Sciences, The University of Texas at Dallas, Richardson, Texas, USA
[2]US BioPharma LLC, Punta Gorda, Florida, USA
[3]Department of Urology, The University of Texas Southwestern Medical Center, Dallas, Texas, USA

## AUTHOR ORCIDs

Philippe E. Zimmern  http://orcid.org/0000-0002-7612-2042
Nicole J. De Nisco  http://orcid.org/0000-0002-7670-5301

## FUNDING

| Funder | Grant(s) | Author(s) |
| --- | --- | --- |
| HHS | NIH | National Institute of Diabetes and Digestive and Kidney Diseases (NIDDK) | 1R01DK131267-01 | Nicole J. De Nisco |

## AUTHOR CONTRIBUTIONS

Namrata V. Sawant, Conceptualization, Data curation, Formal analysis, Investigation, Methodology, Project administration, Validation, Visualization, Writing – original draft, Writing – review and editing | Samuel S. Chang, Data curation, Formal analysis, Investigation, Methodology, Validation, Visualization | Krutika A. Pandit, Investigation | Prachi Khekare, Investigation | W. Randolph Warner, Resources | Philippe E. Zimmern, Conceptualization, Methodology, Resources, Writing – original draft, Writing – review and editing | Nicole J. De Nisco, Conceptualization, Data curation, Formal analysis, Funding acquisition, Methodology, Project administration, Resources, Supervision, Visualization, Writing – original draft, Writing – review and editing

## DATA AVAILABILITY

The data that support the findings of this study are available from the corresponding author upon reasonable request.

## ADDITIONAL FILES

The following material is available online.

Open Peer Review

**PEER REVIEW HISTORY (review-history.pdf).** An accounting of the reviewer comments and feedback.

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
