## [Reviewer comments · Microbiology Spectrum]

Microbiology Spectrum

VesiX Cetylpyridinium Chloride is Rapidly Bactericidal and Reduces Uropathogenic *Escherichia coli* Bladder Epithelial Cell Invasion In Vitro

Namrata Sawant, Samuel Chang, Krutika Pandit, Prachi Khekare, Randolph Warner, Philippe Zimmern, and Nicole De Nisco

Corresponding Author(s): Nicole De Nisco, The University of Texas at Dallas

Review Timeline:

Submission Date:	June 29, 2023
Editorial Decision:	August 27, 2023
Revision Received:	November 4, 2023
Accepted:	December 11, 2023

Editor: Bonnie Prokesch

Reviewer(s): The reviewers have opted to remain anonymous.

Transaction Report:

DOI: <https://doi.org/10.1128/spectrum.02712-23>

August 27, 2023

Dr. Nicole J. De Nisco
The University of Texas at Dallas
Biological Sciences
800 W. Campbell Road
Richardson, Texas 75080

Re: Spectrum02712-23 (VesiX Cetylpyridinium Chloride Solution is Rapidly and Broadly Bactericidal and Reduces Uropathogenic *Escherichia coli* Bladder Epithelial Cell Invasion *In Vitro*)

Dear Dr. Nicole J. De Nisco:

Link Not Available

Sincerely,

Bonnie Prokesch

Journals Department
Reviewer comments:

Reviewer #1 (Comments for the Author):

The manuscript "VesiX Cetylpyridinium Chloride is Rapidly and Broadly Bactericidal and Reduces Uropathogenic *E. coli* Bladder Cell Invasion *In Vitro*" by Sawant et al is a fascinating study on a potential new therapeutic for recurrent UTI. The authors thoroughly tested the efficacy of the VesiX CPC compound both individually and in combination with gentamicin sulfate. The researchers employed several basic microbiology techniques to assess the MIC, MBC, and bactericidal effects of CPC on a variety of different clinical isolates from diverse uropathogenic species. The importance of determining the eukaryotic cellular effects and toxicity was considered and partially experimentally assessed in this work but could be expanded upon. Overall, the authors suggest the potential broad-spectrum use of CPC, in conjunction with gentamicin, to effectively reduce rUTI.

Major Comments:

While the use of CPC seems like a very promising potential therapeutic for UTI, none of the experiments were performed in human urine. The urine is a very diverse and dynamic bodily fluid, and thus, there is great potential for interference with the mechanism of action for this detergent. Likewise, since CPC is targeting the bacterial membrane, the growth conditions of the bacteria can greatly alter hypermucoviscosity, capsule thickness, and other surface components of the bacterial cell that could alter the success of the CPC. This reviewer would like to see Figure 1 repeated in human urine as the medium.

In both Figure 1 & 3, it would be nice to see the incorporation of other quality control and prototype strains for uropathogenic species, in addition to *E. coli*. For example, HI4320 and BB2000 could be used for *Proteus mirabilis*.

Figure 3 should be repeated with other species known to invade the urothelium. Some of these species were mentioned in line 75 of the text.

An additional figure to assess two major bacterial variables, relevant to both antimicrobial activity and UTI biology, should be addressed 1) capsule and 2) biofilm formation. A uronic acid assay and/or Maneval's stain of the different organisms tested in Table 1 (or at minimum, Figure 1) would be relevant for interpreting the effectiveness of CPC. Furthermore, biofilm formation and CAUTI were mentioned several times throughout the manuscript: it would be beneficial to know the capacity of these strains to form biofilms in urine on silicone (or other catheter materials). Testing the effectiveness of this compound on these biofilms would be enlightening for the potential therapeutic use.

Minor Comments:

Perhaps the title can be made more concise.

The introduction ends abruptly.

The limit of detection in Figure 1 should be lowered, if possible. Spread plating 100uL of sample would decrease the limit of detection by 1-log.

Line 191: states the limitation of T24 cells but does not address the limitations of 5637 carcinoma cells, which are known to have overactive pro-survival pathways that may artificially inhibit cell death.

Figures 3B and C appear to be showing the same data. Panel C should be removed, presenting log CFU data as a percentage is not optimal. Also, the deviation and error should still be represented in the relative data, to ensure proper statistical analysis.

Line 232-243: text should be less a rehashing of the results and focus more on why the authors suspect the different resistance and susceptibility patterns among diverse uropathogenic organisms.

McCoy's medium is the optimal choice for T24 cell culture.

Reviewer #2 (Comments for the Author):

Article Title: Vesix Cetylpyridinium Chloride Solution is Rapidly and Broadly Bactericidal and Reduces Uropathogenic Escherichia coli Bladder Epithelial Cell Invasion In Vitro

Journal: Microbiology Spectrum

Manuscript Number: Spectrum02712-23

Concise Summary

This study evaluates the efficacy of Vesix against UPEC responsible for rUTIs in post-menopausal women. In addition, this work evaluates the cytotoxicity of Vesix against cultured bladder epithelial cells. Finally, the authors evaluate the potential utility of combined Vesix and the intravesical antibiotic gentamicin. The findings show that Vesix has a broad bactericidal effect against UPEC organisms with minimal cytotoxic effect on health bladder cells. Finally, the authors determined that gentamicin and Vesix instillations can be used concomitantly.

We commend the efforts put forth by the authors to design and perform this work to determine the bactericidal effects of this treatment in a subset of pathogens, as well as the relative safety of this medication for health bladder cells. We look forward to seeing additional in vivo data regarding the use of Vesix in this difficult-to-treat patient population.

Introduction: This is a very well-written, comprehensive introduction. The authors may consider adding some data about the collateral damage from systemically administered antibiotics (namely, risk of Cdiff) and how bladder instillations are an attractive alternative for many patients who do not want systemic antibiotics to perturb the body's normal flora and microbiome.

Methods:

- Why were bladder cancer cells used versus normal, healthy bladder cells used to evaluate cytotoxicity? Do you think there are differences in how sensitive these cells may be to bacterial infection and exposure to VesiX?

Results: Clear and concise results section, well supported by tables and figures.

Discussion:

- Is there any evidence from the gentamicin literature on what is an acceptable level of cytotoxicity?
- Is there a citation to support this: "While intravesical gentamicin therapy is currently used in patients suffering from rUTI or UTI that are resistant to oral antibiotics, the inability of gentamicin to penetrate epithelial cells may limit its ability to clear intracellular bacteria."?

Staff Comments:

Preparing Revision Guidelines

Please return the manuscript within 60 days; if you cannot complete the modification within this time period, please contact me. If you do not wish to modify the manuscript and prefer to submit it to another journal, please notify me of your decision immediately so that the manuscript may be formally withdrawn from consideration by Microbiology Spectrum.

Response to Reviewers – Sawant et al. 2023

We thank the reviewers for their thoughtful comments and suggestions. We have carefully reviewed the feedback and have added some new data to the manuscript to address reviewer concerns. We have also made all textual changes requested by the reviewers which we believe have improved the manuscript. Please see our point by point response to reviewer comments in blue below.

Reviewer comments:

Reviewer #1 (Comments for the Author):

The manuscript "VesiX Cetylpyridinium Chloride is Rapidly and Broadly Bactericidal and Reduces Uropathogenic E. coli Bladder Cell Invasion In Vitro" by Sawant et al is a fascinating study on a potential new therapeutic for recurrent UTI. The authors thoroughly tested the efficacy of the VesiX CPC compound both individually and in combination with gentamicin sulfate. The researchers employed several basic microbiology techniques to assess the MIC, MBC, and bactericidal effects of CPC on a variety of different clinical isolates from diverse uropathogenic species. The importance of determining the eukaryotic cellular effects and toxicity was considered and partially experimentally assessed in this work but could be expanded upon. Overall, the authors suggest the potential broad-spectrum use of CPC, in conjunction with gentamicin, to effectively reduce rUTI.

Major Comments:

1. While the use of CPC seems like a very promising potential therapeutic for UTI, none of the experiments were performed in human urine. The urine is a very diverse and dynamic bodily fluid, and thus, there is great potential for interference with the mechanism of action for this detergent. Likewise, since CPC is targeting the bacterial membrane, the growth conditions of the bacteria can greatly alter hypermucoviscosity, capsule thickness, and other surface components of the bacterial cell that could alter the success of the CPC. This reviewer would like to see Figure 1 repeated in human urine as the medium.

Thank you for this comment. Yes, the chemical composition of a medium has been shown to impact efficacy. We agree that determining MIC for CPC in urine would be a valuable addition to this work. In response to this, we have performed MIC and MBC experiments for representative strains in pooled human urine supplemented with 30% MHB following a protocol previously published in Cell Reports Medicine (DOI: 10.1016/j.xcrm.2023.101023). We describe these new data in the results section starting on line 161 and have added the data as part of Table 3.

2. In both Figure 1 & 3, it would be nice to see the incorporation of other quality control and prototype strains for uropathogenic species, in addition to E. coli. For example, HI4320 and

BB2000 could be used for *Proteus mirabilis*.

We have now included quality control type strains *E. faecalis* ATCC 29212, *P. mirabilis* HI4320, and *K. pneumoniae* TOP52 in our MIC and MBC assessments and have added these data to Table 3.

3. Figure 3 should be repeated with other species known to invade the urothelium. Some of these species were mentioned in line 75 of the text.

We thank the reviewer for this suggestion and certainly agree that it would be an important future direction for this work. However, we believe it is out of the scope of the current study which is focused on Vesix CPC reducing UPEC urothelial invasion in vitro as suggested by the manuscript title. We have addressed this limitation in the final paragraph of the discussion and highlighted it as an area of future investigation.

4. An additional figure to assess two major bacterial variables, relevant to both antimicrobial activity and UTI biology, should be addressed 1) capsule and 2) biofilm formation. A uronic acid assay and/or Maneval's stain of the different organisms tested in Table 1 (or at minimum, Figure 1) would be relevant for interpreting the effectiveness of CPC. Furthermore, biofilm formation and CAUTI were mentioned several times throughout the manuscript: it would be beneficial to know the capacity of these strains to form biofilms in urine on silicone (or other catheter materials). Testing the effectiveness of this compound on these biofilms would be enlightening for the potential therapeutic use.

We thank the reviewer for these interesting suggestions for research questions that should be addressed in future studies. The goal of this study was to ascertain the antibacterial activity of Vesix CPC and its ability to work with gentamicin to reduce intracellular UPEC in cultured bladder epithelial cells. The suggestions of evaluating capsule and biofilm formation and their correlation with CPC MIC is a very interesting future direction. However, rigorous testing of these hypotheses would require gene deletion and complementation studies that are out of the scope of this work. Even a finding that increased biofilm formation, for example, may be associated with a higher CPC MIC (which itself would require a large number of strains of the same species with varied phenotypes) would be a purely associative observation and defining a causal relationship would require deletion and complementation of genes involved in biofilm formation. We also appreciate the suggestion to evaluate the ability of Vesix CPC on biofilms formed on catheter materials – this again is a fantastic suggestion for a separate study on the efficacy of Vesix in the context of CAUTI. We sincerely thank the reviewer for these excellent suggestions for future studies which we look forward to conducting but respectfully disagree that they are necessary for the conclusions of the current manuscript. We have added a couple sentences to line 262 of the discussion to address the importance of following up on these questions in future work.

Minor Comments:

1. Perhaps the title can be made more concise.

Thank you for this suggestion. We try to be as accurate with our titles as possible such that they faithfully convey the findings of the work. However, we do agree the title is long and have shortened it to:

“VesiX Cetylpyridinium Chloride is Rapidly Bactericidal and Reduces Uropathogenic *Escherichia coli* Bladder Epithelial Cell Invasion In Vitro”

2. The introduction ends abruptly.

The introduction ends with the goals of the current study a brief summary of the major conclusions of the manuscript as is standard for the journal.

3. The limit of detection in Figure 1 should be lowered, if possible. Spread plating 100uL of sample would decrease the limit of detection by 1-log.

This is correct, but the goal of these assays was to determine if VesiX CPC was rapidly bactericidal which requires a 3 log₁₀ reduction in bacterial number as per National Committee for Clinical Laboratory Standards guidelines. Since the starting CFUs were between 10⁷ and 10⁸, a detection limit of 10³ is more than sufficient to determine if a 3 log₁₀ reduction occurred.

4. Line 191: states the limitation of T24 cells but does not address the limitations of 5637 carcinoma cells, which are known to have overactive pro-survival pathways that may artificially inhibit cell death.

Thank you for pointing this out. We have added the following text to the Discussion to address the limitations of using both T24 and 5637 cancer cell lines:

“A limitation of these findings is that 5637 cells have mutations in TP53, RB1 and ERBB2 genes, while T24 cells also have mutated TP53 and HRAS genes. These oncogenic alterations are known to activate pro-survival pathways, inhibiting cell death and could potentially impact the accurate prediction of cytotoxicity. To overcome this limitation, additional studies with normal bladder epithelial cells could be performed to further assess the cytotoxic effect of VesiX CPC.”

5. Figures 3B and C appear to be showing the same data. Panel C should be removed, presenting log CFU data as a percentage is not optimal. Also, the deviation and error should still be represented in the relative data, to ensure proper statistical analysis.

We agree that the statistical test is redundant since it is the same data and have removed this analysis from 3C. We have left 3C to aid in reader interpretation as it makes it easier to see the difference in invasion frequency. But all conclusions are now only based on the CFU data in Figure 3B.

6. Line 232-243: text should be less a rehashing of the results and focus more on why the authors suspect the different resistance and susceptibility patterns among diverse uropathogenic organisms.

Thank you for this comment, we have added text to this section to discuss the possible contribution of outer membrane modification, capsule and biofilm to increased VesiX CPC

resistance or tolerance. We have also removed a few phrases and extra words in this section and the following sentences to reduce the amount of rehashing:

“Overall, VesiX formulations with CPC concentration of 0.0063% and 0.0125% were bactericidal (MBC/MIC ratio ≤ 4) for the majority of tested uropathogens.”

“Foremost, we ensured CPC and gentamicin could be used in combination without having an antagonistic effect on the individual effectiveness of each antimicrobial agent.”

7. McCoy's medium is the optimal choice for T24 cell culture.

Thank you for this comment, we used RPMI so that the culture media would be standard between experiments. We did not have any difficulty cultivating T24 cells in RPMI. We have added our rationale for this choice in the results section.

Reviewer #2 (Comments for the Author):

Article Title: VesiX Cetylpyridinium Chloride Solution is Rapidly and Broadly Bactericidal and Reduces Uropathogenic Escherichia coli Bladder Epithelial Cell Invasion In Vitro

Journal: Microbiology Spectrum

Manuscript Number: Spectrum02712-23

Concise Summary

This is a study evaluates the efficacy VesiX against UPEC responsible for rUTIs in post-menopausal women. In addition, this work evaluates the cytotoxicity of VesiX against cultured bladder epithelial cells. Finally, the authors evaluate the potential utility of combined VesiX and the intravesical antibiotic gentamicin. The findings show that VesiX has a broad bactericidal effect against UPEC organisms with minimal cytotoxic effect on health bladder cells. Finally, the determined that gentamicin and VesiX instillations can be used concomitantly.

We commend the efforts put for by the authors to design and perform this work to determine the bactericidal effects of this treatment in a subset of pathogens, as well of the relative safety of this medication for health bladder cells. We look forward to seeing additional in vivo data regarding the use of VesiX in this difficult to treat patient population.

1. Introduction: This is a very well-written, comprehensive introduction. The authors may consider adding some data about the collateral damage from systemically administered antibiotics (namely, risk of Cdiff) and how bladder instillations are an attractive alternative for many patients who do not want systemic antibiotics to perturb the body's normal flora and microbiome.

We thank the reviewer for this comment and suggestion to add more data about the collateral damage of systemic antibiotics. We have added a sentence about this and a reference to line 72 in the introduction.

2. Methods:

- Why were bladder cancer cells used versus normal, healthy bladder cells used to evaluate cytotoxicity? Do you think there are differences in how sensitive these cells may be to bacterial infection and exposure to Vesix?

We used the 5637 and T24 cell lines as opposed to normal, healthy bladder cells because the ability of UPEC to invade these cell lines has been documented and they are highly feasible to culture for these experiments. Primary bladder epithelial cells (BdECs) are available on ATCC, but they require specialized medium and are very difficult to culture and maintain. Because these cells are not differentiated, the added benefit of using them was unclear. We recognize the limitation of our use of 5637 and T24 cells and have added discussion of this in the second paragraph of the discussion.

3. Results: Clear and concise results section, well supported by tables and figures.

Thank you for this comment.

4. Discussion:

- Is there any evidence from the gentamicin literature on what is an acceptable level of cytotoxicity?

Thank you for pointing that out. The acceptable level of cytotoxicity was determined as per ISO 10993-5:2009. We have added a sentence about this in line 433 in the materials and methods.

- Is there a citation to support this: "While intravesical gentamicin therapy is currently used in patients suffering from rUTI or UTI that are resistant to oral antibiotics, the inability of gentamicin to penetrate epithelial cells may limit its ability to clear intracellular bacteria."?

Yes, thank you for catching this oversight. We have added the appropriate reference for this statement.

Re: Spectrum02712-23R1 (VesiX Cetylpyridinium Chloride is Rapidly Bactericidal and Reduces Uropathogenic *Escherichia coli* Bladder Epithelial Cell Invasion In Vitro)

Dear Dr. Nicole J. De Nisco:

Your manuscript has been accepted, and I am forwarding it to the ASM production staff for publication. Your paper will first be checked to make sure all elements meet the technical requirements. ASM staff will contact you if anything needs to be revised before copyediting and production can begin. Otherwise, you will be notified when your proofs are ready to be viewed.

Sincerely,
Bonnie Prokesch
Editor
Microbiology Spectrum

Reviewer #1 (Comments for the Author):

I agree with the rebuttal to Major comment #3, testing other species is out of scope. However, I do think that other UPEC strains should be tested. The MIC and MCB have already been completed with four additional clinical UPEC isolates. This is important because we know there is extreme variability in the intracellular invasion capacity of different UPEC strains (PMID: 28330863, among others).

While I respect and appreciate the response to previous Major concern #4, I still believe better methodological rigor could be applied. I now understand and agree that work to define the mechanism of action is out of the scope for this manuscript. However, I do believe that the reduction in invasion (Figure 3) should be shown with an additional complementary technique. For example, microscopy, to visualize this phenotype.

All other concerns were sufficiently addressed.